# Association between Mortality and Short-Term Exposure to Particles, Ozone and Nitrogen Dioxide in Stockholm, Sweden

**DOI:** 10.3390/ijerph16061028

**Published:** 2019-03-21

**Authors:** Henrik Olstrup, Christer Johansson, Bertil Forsberg, Christofer Åström

**Affiliations:** 1Atmospheric Science Unit, Department of Environmental Science and Analytical Chemistry, Stockholm University, 11418 Stockholm, Sweden; christer.johansson@aces.su.se; 2Environment and Health Administration, SLB, Box 8136, 104 20 Stockholm, Sweden; 3Section of Sustainable Health, Department of Public Health and Clinical Medicine, Umeå University, 90187 Umeå, Sweden; bertil.forsberg@umu.se (B.F.); christofer.astrom@umu.se (C.Å.)

**Keywords:** particle number count (PNC), PM_2.5–10_, exposure, mortality, ozone, excess risk, linear regression

## Abstract

In this study, the effects on daily mortality in Stockholm associated with short-term exposure to ultrafine particles (measured as number of particles with a diameter larger than 4 nm, PNC_4_), black carbon (BC) and coarse particles (PM_2.5–10_) have been compared with the effects from more common traffic-pollution indicators (PM_10_, PM_2.5_ and NO_2_) and O_3_ during the period 2000–2016. Air pollution exposure was estimated from measurements at a 20 m high building in central Stockholm. The associations between daily mortality lagged up to two days (lag 02) and the different air pollutants were modelled by using Poisson regression. The pollutants with the strongest indications of an independent effect on daily mortality were O_3_, PM_2.5–10_ and PM_10_. In the single-pollutant model, an interquartile range (IQR) increase in O_3_ was associated with an increase in daily mortality of 2.0% (95% CI: 1.1–3.0) for lag 01 and 1.9% (95% CI: 1.0–2.9) for lag 02. An IQR increase in PM_2.5–10_ was associated with an increase in daily mortality of 0.8% (95% CI: 0.1–1.5) for lag 01 and 1.1% (95% CI: 0.4–1.8) for lag 02. PM_10_ was associated with a significant increase only at lag 02, with 0.8% (95% CI: 0.08–1.4) increase in daily mortality associated with an IQR increase in the concentration. NO_2_ exhibits negative associations with mortality. The significant excess risk associated with O_3_ remained significant in two-pollutant models after adjustments for PM_2.5–10_, BC and NO_2_. The significant excess risk associated with PM_2.5–10_ remained significant in a two-pollutant model after adjustment for NO_2_. The significantly negative associations for NO_2_ remained significant in two-pollutant models after adjustments for PM_2.5–10_, O_3_ and BC. A potential reason for these findings, where statistically significant excess risks were found for O_3_, PM_2.5–10_ and PM_10_, but not for NO_2_, PM_2.5_, PNC_4_ and BC, is behavioral factors that lead to misclassification in the exposure. The concentrations of O_3_ and PM_2.5–10_ are in general highest during sunny and dry days during the spring, when exposure to outdoor air tend to increase, while the opposite applies to NO_2_, PNC_4_ and BC, with the highest concentrations during the short winter days with cold weather, when people are less exposed to outdoor air.

## 1. Introduction

For PM_10,_ PM_2.5_, NO_2_ and O_3_, the short-term health effects in terms of increased daily mortality have been investigated in many studies. Results from original studies have also been combined in meta-analyzes. For NO_2_, a significant association between short-term exposure and mortality, based on 60 studies from different parts of the world, was described in Mills et al. [1]. In this study, two-pollutant models showed that the association between daily mortality and exposure to NO_2_ was largely independent of PM mass. Bell et al. [2] performed a meta-analysis of the short-term health effects related to exposure to O_3_, based on 144 effect estimates from 39 time-series studies from different parts of the world, providing evidence of an association between O_3_ exposure and daily mortality. A systematic review and a meta-analysis of the short-term health effects related to PM_2.5_ and PM_10_ in the Chinese population was performed by Lu et al. [3]. In this study, the excess risks of non-accidental mortality, cardiovascular mortality and respiratory mortality exhibited statistically significant (95% CI) risk increases for exposure to both PM_2.5_ and PM_10_. 

In contrast to the above mentioned air pollutants, when considering the health effects associated with exposure to ultrafine particles (UFP), BC and coarse particles, there are relatively few epidemiological studies that have investigated the short-term effects on daily mortality. In a meta-analysis by Atkinson et al. [4], the majority of the associations between short-term exposure to PNC and mortality were positive but non-significant. In this meta-analysis, the included studies used different size-distributions and modes, varying in the range of 3 nm to 0.49 μm [4]. In Samoli et al. [5], the daily mortality in London associated with exposure to an IQR increase of four different size-distributions of ultrafine particles (diameter < 0.6 μm) showed no positive significant associations. The UFIREG study from Central Europe [6] found positive but non-significant associations between short-term respiratory mortality and exposure to UFP (20–100 nm). Also the UF & Health study [7], including Nordic, Central European and Mediterranean cities, found weak (non-significant) evidence of an association between UFP (an increase in 10,000 particles cm^−3^) and total and cause-specific daily mortality. In Chen et al. [8], pooled associations between UFP (particle number count in the ultrafine range (≤100 nm), or total particle number count ≤3000 nm, as a proxy for UFP) and total and cardiovascular mortality were overall positive, and generally stronger at high compared to low air temperatures. On days with high air temperatures (>75th percentile), an increase in 10,000 particles cm^−3^ was associated with a significant increase in cardiovascular mortality [8]. A study including three Spanish cities found positive but non-significant associations between primary UFP and total daily mortality in Barcelona and Tenerife, while in Huelva, an association between total daily mortality and secondary UFP was found [9]. In the Ruhr Area in Germany, size-specific PNC (PNC_<100_ and PNC_100–750_) was for some specific lags significantly associated with daily mortality, and with the strongest positive associations between PNC_100–750_ and natural daily mortality [10]. To sum up, research has not been able to conclusively establish a link between short-term UFP exposure and daily mortality. These uncertain results might be attributed to the different origins of UFP in the cities involved in the studies, or even to the different origin of UFP for different days in a city [11].

For particles in the coarse fraction (PM_2.5–10_), the health effects are less investigated in comparison with both PM_2.5_ and PM_10_ and UFP. Particles in the coarse fraction (2.5–10 µm in diameter) are mainly formed by mechanical grinding and resuspension of solid materials. In the cities, these particles can consist of both crustal and organic materials from the asphalt, rubber from tires and metals from studded tires and brakes. The short-term effects on mortality associated with exposure to PM_2.5–10_ have been analyzed in a review study from 2005 [12]. Some of the studies that were analyzed showed statistically significant associations between short-term mortality effects and a 10 µg m^−3^ increase in PM_2.5–10_. The effects were largest in studies from arid regions, where the PM_2.5–10_ concentrations were relatively high. However, for long-term studies on mortality effects, no associations were found for PM_2.5–10_. In another systemic review and meta-analysis from 2014 [13], focusing on the health effects associated with exposure to PM_2.5–10_, significant associations between short-term exposure to PM_2.5–10_ and mortality were found, with more robust relationships for respiratory compared to cardiovascular endpoints. The associations were highly heterogeneous, but with differences related to geographic region and the average PM_2.5–10_ concentrations. After adjustments for publication bias and PM_2.5_ exposure, the effects associated with PM_2.5–10_ became weaker and less precise, but the positive effects remained for short-term exposure. For long-term exposure associated with PM_2.5–10_, evidence of increased mortality was found, but after adjustment for PM_2.5_ exposure, these associations were non-significant [13].

The purpose of this study was to compare the excess risks of daily mortality associated with potentially relevant traffic pollution indicators, namely UFP (PNC_4_), BC and PM_2.5–10_, and compare these results with the excess risks of the more established indicators, namely PM_10_, PM_2.5_ and NO_2_, as well as with O_3_. We have examined the associations between daily mortality and exposure to NO_2_, O_3_, PM_2.5_, PM_10_, PM_2.5–10_, PNC_4_ and BC, based on data from an urban background measuring station in Stockholm during the period 2000–2016.

## 2. Materials and Methods

This study covers the city of Stockholm with a population of approximately 0.8–0.9 million people over the period 2000–2016. Population data were obtained from the Swedish Central Bureau of Statistics. Natural cause mortality data were obtained from the National Cause of Death Register. Natural cause mortality was defined on the basis of the underlying cause of death (ICD-10: A00–R99), and these data included the daily number of deaths from natural causes occurring among the registered population. 

Air pollution exposure was estimated from central measurements made on the roof-top of a 20 m high building in central Stockholm. The monitoring station is part of the city’s regulatory air pollution control network, and equipped with reference (or equivalent) instruments for regulated pollutants according to the EU air quality directive for NO_2_, O_3_, PM_2.5_ and PM_10_ (Table A1). The O_3_ measurements are based on daily maximum 8-h mean. In addition, it includes measurements of unregulated black carbon (BC), total particle number concentrations (PNC_4_), and the coarse fraction (PM_2.5–10_), estimated by subtracting PM_2.5_ from PM_10_. Different instruments have been used to measure BC. However, when the hourly mean values of two different BC measurements techniques in Stockholm were compared in 2006, the R-values were 0.87 and 0.95, respectively, at the two measurement sites [14]. For the measurements of ultrafine particles (PNC_4_ and PNC_7_), an instrument was used that registered all particles larger than 7 nm (PNC_7_) during the period from May 2001 until November 2013. From March 2008 to the end of the period (December 2016), another instrument was used to record particles larger than 4 nm (PNC_4_). By applying a linear regression between PNC_4_ and PNC_7_ during overlapping periods, measured PNC_7_ was used to construct a more complete time-series of PNC_4_. The mean value of the measured concentrations of PNC_4_ was 8650 cm^−3^, with a root mean square error (RMSE) of 1355 cm^−3^ in relation to the modeled PNC_4_ values, based on the linear regression with PNC_7_ (see Figure A2 in Appendix A). Several plots illustrating the daily and monthly mean concentrations during 2000–2016, and the correlations between different pairs of the analyzed air pollutants, have been performed by using the “Openair” package [15]. 

The associations between different air pollutants and the daily mortality were modelled by using a quasi-Poisson regression model with a logistic link function. The model estimated the effect of an IQR increase of air pollutants on daily mortality for lag 01 and 02, while controlling for other time-varying factors by assuming a linear additive effect on a logarithmic scale:Log(Y*i*) = Intercept + f(AP*i*) + f(W*i*) + DOW*i* + (long-term trend)(1)
where AP*i* is the concentration of a specific or a combination of air pollutants on day *i*, W*i* is variables controlling for the weather on day *i*, more specifically maximum temperature and snowfall, DOW*i* is the day of week, and the long-time trend is a smooth function varying over time to capture any long-term and seasonal patterns in mortality. The smooth function used was a penalized regression spline restricted to 5 d.f. (degrees of freedom) per year. Snowfall has been included, since it is a risk factor for daily mortality, as described in Auger et al. (2017) [16]. All pollutants were modelled by assuming a linear relationship with daily mortality. Air pollutants were first modelled in single-pollutant models (Figure 4), and traffic-related pollutants with effect estimates with a *p*-value smaller than 0.2 were included in multi-pollutant models together with O_3_ (Figures 5–7). We also investigated the correlation matrix (Figure 3), and included pollutants that were negatively correlated, or positively correlated, but with opposite effects in the single-pollutant model (Figure 4). 

Temperature were adjusted for by using two different smooth functions corresponding to the different lag-windows of 0–2 and 3–10. The model allowed for the use of 4 d.f. for each function. In the sensitivity analysis, we added models allowing for 8 d.f. in the smooth temperature functions as well as adding another temperature variable with temperatures from lag 11–20. In addition, we used an indicator variable to identify if the modelled PNC_4_ data (see Figure A2) generated different risk estimates in comparison with the measured data.

## 3. Results

### 3.1. Descriptive Data 

The descriptive data are presented in Table 1. An overview of the temporal variation of the daily mean concentrations of air pollutants and daily mortality in Stockholm is given in Figure A1 in Appendix A. For most pollutants, data capture is high, >80% (Table 1); the exceptions are BC (53%) and PNC (70%). 

There were also pronounced seasonal variations in the concentrations, as shown in Figure 1 and Figure 2. NO_2_, PNC and BC exhibited the highest concentrations during winter (October to March) and lowest in summer (June–July), whereas PM_10_, PM_2.5_, PM_2.5–10_ and O_3_ exhibited peak concentrations during late winter to early summer (March–May).

Figure 3 shows the correlations between all pairs of data for the measured air pollutants. High correlations (R > 0.6) were found between pairs of NO_2_, PNC_7_, PNC_4_ and PNC, reflecting their common origins. Likewise the pairs PM_10_-PM_2.5_, PM_10_-PM_2_._5–10_ and BC-PM_2.5_ showed high correlations. O_3_ showed negative correlations with NO_2_, PNC and BC. Note that the R-values in Figure 3 are given in percent.

### 3.2. The Calculated Excess Risks

Figure 4, Figure 5, Figure 6 and Figure 7 show the calculated excess risks for daily mortality associated with an IQR increase of the measured pollutants in Stockholm during 2000–2016. The IQR values for the measured pollutants are presented in Table 1. Figure 4 shows single-pollutant models, where the excess risks for lag 01 and 02 associated with an IQR increase of the different pollutants are presented. Lag 01 and 02 represent a lagging effect of the same and the previous day, and the same and the previous two days, respectively. Figure 5, Figure 6 and Figure 7 show multi-pollutant models, where the effects of the modeled pollutants are adjusted for each other. All multi-pollutant models are based on lag 02. 

In the single-pollutant model (Figure 4), O_3_ exhibits significant excess risks of 2.0% (95% CI: 1.1–3.0) for lag 01 and 1.9% (95% CI: 1.0–2.9) for lag 02 associated with an IQR increase in concentration. PM_2.5–10_ exhibits significant excess risks of 0.8% (95% CI: 0.1–1.5) for lag 01 and 1.1% (95% CI: 0.4–1.8) for lag 02 associated with an IQR increase in concentration. PM_10_ exhibits a significant excess risk of 0.8% (95% CI: 0.08–1.4) for lag 02 associated with an IQR increase in concentration. NO_2_ exhibits negative risks of −1.6% (95% CI: −0.6–−2.7) for lag 01 and −1.5% (95% CI: −0.5–−2.5) for lag 02 associated with an IQR increase in concentration. The other risk estimates in the single-pollutant model are not statistically significant. 

In the multi-pollutant models (Figure 5, Figure 6 and Figure 7), the risk estimates in the single-pollutant model have been adjusted for the effects of some other pollutants. Pollutants that were negatively correlated (see Figure 3), or positively correlated, but with opposite effects in the single-pollutant model (Figure 4) have been used in the multi-pollutant models. For O_3_, the significant effect for lag02 in the single-pollutant model remained significant in two-pollutant models after adjustments for PM_2.5–10_, BC and NO_2_ (Figure 4, Figure 5, Figure 6 and Figure 7). For PM_2.5–10_, the significant effect for lag02 in the single-pollutant model remained significant in the two-pollutant after adjustment for NO_2_, and also after adjustments for both O_3_ and NO_2_ together (Figure 5). The significantly negative effect associated with NO_2_ for lag02 in the single-pollutant model remained significant in two-pollutant models after adjustments for PM_2.5–10_, BC and O_3_ (Figure 5, Figure 6 and Figure 7). The significantly negative effect associated with NO_2_ also remained significant after adjustment for both PM_2.5–10_ and O_3_ together (Figure 7). Modelling the effect of PNC_4_, while adjusting for O_3_, increased the effect estimate by 23%, while remaining non-significant. The estimated effect of PM_2.5_ was to some degree affected by the introduction of NO_2_ in the model, where the negative, non-significant estimate for PM_2.5_ (*p*-value: 0.95) changed to a small positive effect, while remaining non-significant (*p*-value: 0.65).

### 3.3. Sensitivity Analysis

In the sensitivity analysis, we allowed for more flexible temperature associations. Increasing the number of d.f. allowed in the temperature function lowered the *p*-values for mainly PM_10_ and PM_2.5_. However, adding a temperature variable, adjusting for a longer time-frame of 11–20 days prior, rendered the PM variables insignificant. The other air pollution variables were essentially unaffected by inclusion of a longer temperature adjustment. The reasons for these patterns are unknown, but resuspension of road dust depends on complex changes in weather conditions, e.g., from periods with rain or snow to dry weather, which could also affect mortality.

The investigation of the modelled PNC_4_ data (Figure A2) found that there was no difference between the estimates generated by the modelled and the measured data. Consequently, the lack of statistically significant excess risks associated with PNC_4_ is not caused by the use of the modelled PNC_4_ data, based on the linear regression with PNC_7_ (Figure A2). 

## 4. Discussion

### 4.1. Local and Non-Local Sources

This study includes pollutants that are relatively good indicators of different local and non-local sources. NO_2_, PNC are all mainly influenced by local vehicle exhaust emissions [17,18], and are therefore highly correlated, as shown in Figure 3. BC is also emitted mainly from local vehicle exhaust, but is also influenced by long-range transport [14], making the correlation with NO_2_ and PNC somewhat smaller. The coarse particle fraction, PM_2.5–10_, is mainly due to local road-dust suspension [17], and since PM_10_ largely consists of coarse particles, it is highly correlated with PM_2.5–10_. PM_2.5_ is dominated by long-range transported secondary particles, and shows low correlations with pollutants like NO_2_, PNC and PM_2.5–10_, mainly influenced by local sources. However, PM_2.5_ shows higher correlations with BC and PM_10_ due to some influence of long-range transport on these compounds. And finally, the O_3_ concentrations in the city depend mainly on the long-distance transport, but are to some degree also influenced by the chemical reactions involving nitrogen monoxide (NO). The O_3_ concentrations are therefore reduced when primary exhaust concentrations are high during stagnant conditions.

It should be noted that the influence on PM_10_ of local vehicle generated road-dust suspension is relatively large in Stockholm compared to many other cities in Europe. Even though also NO_2_ and PNC originate from local road traffic, the temporal correlations with road dust is very low (Figure 3). The reason for this is that road-dust suspension is highly influenced by the wetness of the road surfaces, as shown earlier by Johansson et al. [17]. So, in summary, the mix of pollutants included in this study are indicators of local road traffic emissions from vehicle exhaust and local non-exhaust particles, non-local secondary particulate matter and photochemical pollutants (O_3_).

### 4.2. Representativeness of One Central Monitoring Station for Population Exposure

Clearly, the possibility to quantify any associations between mortality and different pollutants depends on how well the exposure can be quantified. In this study, we have used one single urban background site, assuming that the temporal variability at this site reflects the temporal variability of the exposure in the population. A high temporal correlation between ambient concentrations at different measuring sites within a city means that one centrally located measuring station should be enough in order to estimate the short-term variations in pollutant concentrations in time-series studies, even though it is inadequate when it comes to estimate the long-term health effects, due to the spatial gradients in exposure concentrations [17,19]. 

Since PM_2.5_ are mainly influenced by non-local sources, spatial variations in ambient concentrations are small, and temporal variations will be very similar everywhere in the city. This has also been verified earlier for Stockholm in the TRAPCA study by Cyrys et al. [20], and it is also well known from other studies (see e.g., review by Monn [21]). O_3_ concentrations in Stockholm are also mainly influenced by long-range transport, even though there is some impact of the titration by NO_x_ (photochemical removal) close to densely trafficked roads with high emissions of NO_x_. There is no local photochemical production of O_3_ in Stockholm. 

For pollutants like PNC, BC and NO_2_, with road traffic emissions being the main source, the temporal variability in the concentrations may be expected to be quite similar everywhere in the city, as traffic intensities usually show similar temporal variations along different roads in the city. High temporal correlations (R ≈ 0.8) between traffic sites have been observed for PNC in Helsinki (Buzorius et al. [22]), and between 24-h mean concentrations of PNC at central sites and homes in Amsterdam, Athens, Birmingham and Helsinki, with a median correlation for PNC per city in the range of 0.67 and 0.76, as shown in Puustinen et al. [23]. Likewise, Cyrys et al. [19], found high correlations (R > 0.80) for PNC when they compared four different measurement stations in Augsburg. However, the variability of the concentrations of PNC in different cities may not be driven by the same emissions sources and atmospheric processes, and the PNC variability does not always indicate the impact of road traffic on air quality [24]. The variability in PNC depends on meteorology and on the size of the smallest particles considered due to the increasing influence of particle dynamics on particles smaller than 20 nm. Model calculations of PNC in Stockholm by Gidhagen et al. [18] showed that episodic losses of nanoparticles due to coagulation and dry deposition, some kilometers downwind of major roads, rise in connection with low wind speed and suppressed turbulent mixing. Similar results was found by Karl et al. [25], based on model calculations of PNC in Oslo, Helsinki and Rotterdam. Removal due to coagulation and deposition may thus lead to different temporal variations in different parts of a city. Moreover, for particles smaller than 100 nm, the formation is highly temperature dependent, as has been shown for Stockholm, where the number of particles, normalized by NO_x_, increases with decreasing temperature [26]. In addition, exposures in microenvironments, like indoors, might be very important for the daily exposure doses of PNC (Kumar et al. [27]). And for BC, the spatiotemporal variability was not found to be very high in Stockholm; different urban sites were poorly correlated even for daily averages (R < 0.70) [14], indicating that a single central measurement site would lead to misclassifications in the exposure. 

Consequently, this means that the central monitor should reflect day-to-day variations in the exposure to ambient PM_2.5_ and O_3_, but may be less good for BC and PNC.

### 4.3. The Estimated Excess Risks and Explanatory Factors

In both epidemiological and clinical studies, short-term exposure to air pollutants has been demonstrated to increase the mortality related to cardiovascular and respiratory diseases. Regarding short-term mortality related to PM, there are several potential biological mechanisms behind this relationship. Exposure to PM in both the coarse (PM_2.5–10_) and the fine (PM_2.5_) fraction induces oxidative stress and inflammation. Exposure to PM_2.5_ can also affect the autonomic nervous system, and can thereby cause alterations in the autonomic control of the heart, which is also a risk factor for cardiovascular mortality [28].

The results presented in the single-pollutant model in Figure 4 show statistically significant positive excess risks for O_3_ and PM_2.5–10_ and PM_10_, but with no positive significant excess risks associated with exposure to the other pollutants. However, when comparing these coefficients with the results from similar studies, there are similarities regarding the results. In Meister et al. [29], a stronger effect on daily mortality (lag01) from PM_2.5–10_ in comparison with PM_2.5_ was found for Stockholm, based on data from 2000 to 2008. 

In this study, statistically significant excess risks for O_3_ were found for both lag 01 and lag 02 in the single-pollutant model, and the excess risks remained significant in all two-pollutant models. However, this phenomenon is in line with other studies that have done similar analyzes. In Raza et al. [30], where short-term effects of air pollution on out-of-hospital cardiac arrest in Stockholm were analyzed, significant effects were observed for O_3_, but not for PM_2.5_, PM_2.5–10_, NO_x_ and NO_2_. In another study performed in Stockholm, where short-term exposure to ozone and mortality in subjects with and without previous cardiovascular disease was analyzed, significant associations were found for O_3_, and these associations remained basically unchanged in two-pollutant models with NO_2_ and PM_10_ [31]. 

The results in Figure 4 exhibit a tangible pattern, with significant excess risks for O_3_, PM_2.5–10_ and PM_10_, but with non-significant excess risks for all the exhaust-related pollutants. In Stockholm, the mass of PM_10_ consists largely of mechanically generated particles from road abrasion, and since PM_10_ and PM_2.5–10_ are highly correlated (R = 0.81), their sources are thus also largely the same. 

There are some possible reasons for the robust associations between exposure to O_3_, PM_2.5–10_ and PM_10_, and the daily mortality. Besides the established harmful effects, one possible reason may be that the measurement data for O_3_, PM_2.5–10_ and PM_10_ reflect the exposure better than for the other pollutants due to behavioral factors. Increased O_3_ concentrations coincides with sunny and warm days in spring and summer when people spend more time outdoors and allow windows to be opened to a greater extent. This will increase the exposure and thereby contribute to the significant excess risks. The concentrations of PM_2.5–10_ and PM_10_ also tend to increase during sunny days in spring and early summer due to suspension from dry roadways, which will then cause a higher exposure during these days. Contrariwise, the exhaust-related emissions tend to be lowest when the outdoor activities tend to be highest and vice versa, which possibly can contribute to the absence of significant positive associations for all exhaust-related pollutants (NO_2_, PM_2.5_, PNC_4_ and BC) in Figure 4. In this way, the exposure misclassification may work in different directions for different pollutants. The uncommon result for NO_2_ could also partly be explained by the high contribution from local sources resulting in larger weather influences. Large spatial variation in combination with the use of one centrally located measuring station may also increase the exposure misclassification for NO_2_, if the average exposure levels vary differently than at the central monitoring station. Negative associations between NO_2_ and cardiovascular mortality were also observed for the northernmost cities of Stockholm and Helsinki in the APHEA2 Study [32]. Even though the problem with behavioral factors has been discussed also for southern European cities (Chiusolo et al. [33]), it is probably not as pronounced as in the northern European cities, where the contrast in weather between summer and winter is much more pronounced, which can contribute to the negative associations observed in Stockholm and Helsinki. 

### 4.4. Strengths and Limitations of This Study 

A strength of this study is that we have an extensive dataset with continuous measurements of many pollutants that are good indicators of different sources: local vehicle exhaust and non-exhaust emissions, non-local secondary particles and an important gaseous oxidant (O_3_). Another strength is that we use high quality mortality data from the National Cause of Death Register. 

The most obvious limitation of this study is that the measured concentrations are obtained from one centrally located measuring station, which potentially creates exposure misclassifications among the population. Another limitation of the study is the poor data capture for PNC_4_ and PNC_7_, where a simple linear regression during overlapping periods has been used to construct a complete time series of PNC_4_. However, the calculated RMSE between the measured and the modeled PNC_4_ concentrations is relatively small in comparison with the mean value of PNC_4_ during this overlapping period, and the investigation of the modelled PNC_4_ data found that there was no difference between the estimates generated by the modelled and the measured data. Another limitation is that we have used particle counters that include nanoparticles, which may be subject to large variability due to dynamic processes like coagulation and condensation/evaporation, and also due to dry deposition. By including only particles larger than e.g., 20 nm, the influence of particle dynamics and deposition may be avoided and make the exposure estimate using a single site more representative for the population. 

## 5. Conclusions

The conclusion of this study is that the excess risks associated with exposure to exhaust emissions (NO_2_, PM_2.5_, BC and PNC_4_) exhibit much more uncertain relationships in comparison with O_3_, PM_2.5–10_ and PM_10_. The results, where significant associations were found only for O_3_, PM_2.5–10_ and PM_10_ are, however, in line with other studies from Stockholm which have analyzed similar relationships. The spatial and temporal variations associated with pollutants of local origin can make it harder to estimate population exposure based on one centrally located measuring station. However, the potential reason for the result findings, where statistically significant positive excess risks were found for O_3_, PM_2.5–10_ and PM_10_, but not for NO_2_, PM_2.5_, PNC_4_ and BC, is probably to a large part caused by behavioral factors. The concentrations of O_3_ and PM_2.5–10_ are in general high during sunny days, when outdoor activities tend to increase, while the opposite applies to NO_2_, PM_2.5_, PNC_4_ and BC, with the highest concentrations during the winter months.

## Figures and Tables

**Figure 1 ijerph-16-01028-f001:**
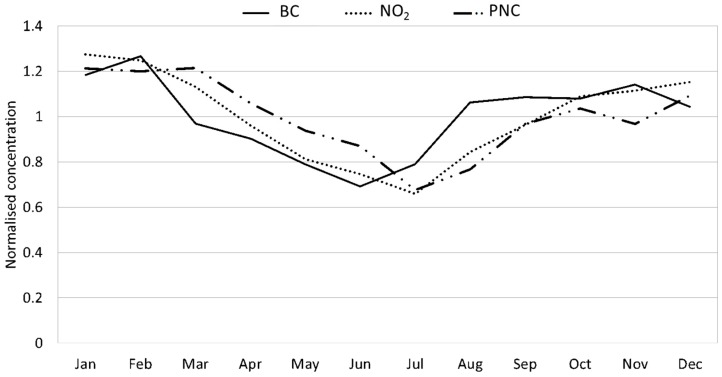
Normalized monthly mean concentrations of BC, NO_2_ and PNC for the measurements in central Stockholm during 2000–2016.

**Figure 2 ijerph-16-01028-f002:**
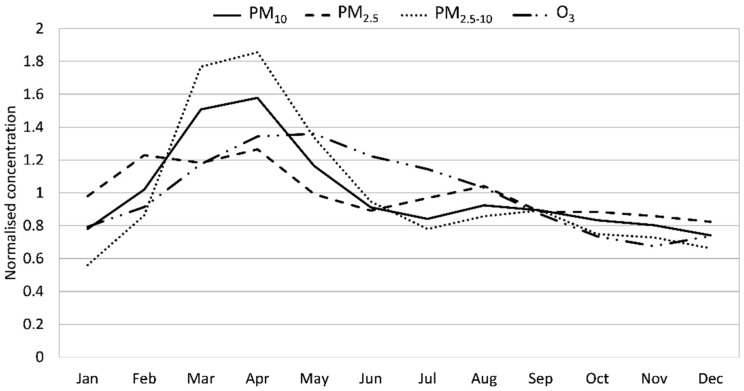
Normalized monthly mean concentrations of PM_10_, PM_2.5_, PM_2.5–10_ and O_3_ for the measurements in central Stockholm during 2000–2016.

**Figure 3 ijerph-16-01028-f003:**
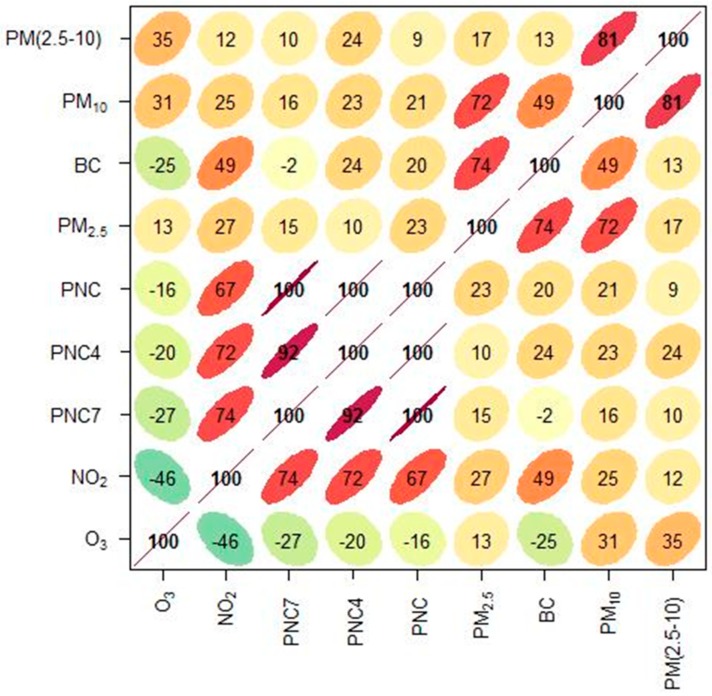
Correlation matrix showing pairwise Pearson correlation coefficients as percentage values. The correlations are illustrated by shapes (ellipses), colors and the numeric value. The ellipses are visual representations of the scatter plots. With a perfect positive correlation, a line at 45 degrees positive slope is drawn. For zero correlation, the shape becomes a circle.

**Figure 4 ijerph-16-01028-f004:**
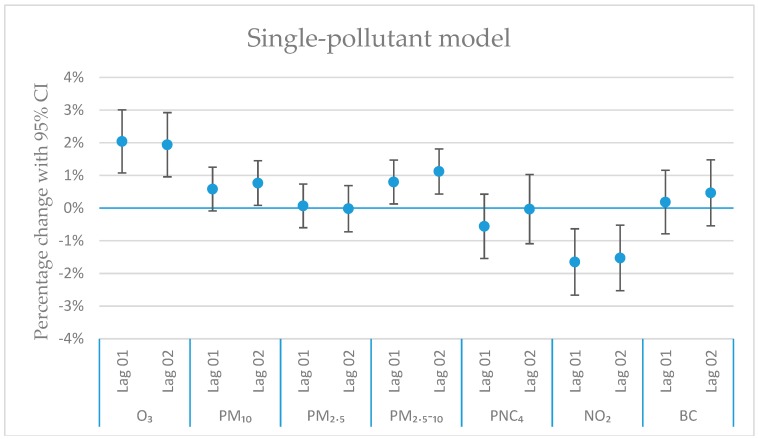
The estimated change in daily mortality (with 95% CI) for an IQR increase in concentration (lag01 and lag02) to the different air pollutants in a single-pollutant model.

**Figure 5 ijerph-16-01028-f005:**
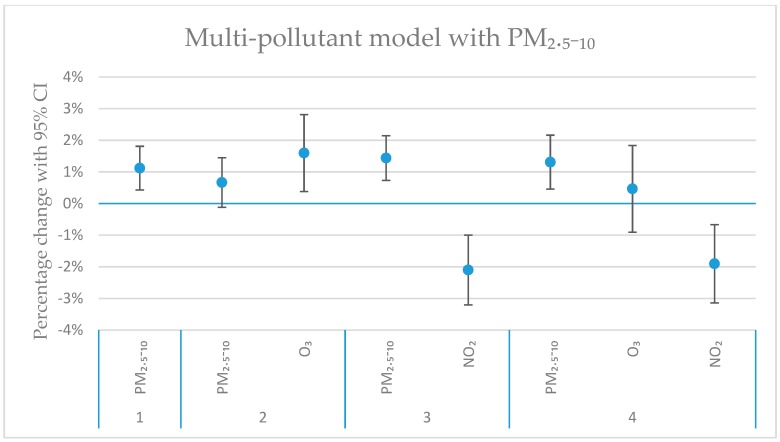
The estimated change in daily mortality (with 95% CI) for an IQR increase in concentration (lag02) to PM_2.5–10_, O_3_ and NO_2_ in a multi-pollutant model. Model 1 (furthest to the left) represents the single-pollutant estimate of PM_2.5–10_. Model 2 and 3 represent two-pollutant models, with O_3_ (Model 2), and with NO_2_ (Model 3). In Model 4, both PM_2.5–10_, O_3_ and NO_2_ are included.

**Figure 6 ijerph-16-01028-f006:**
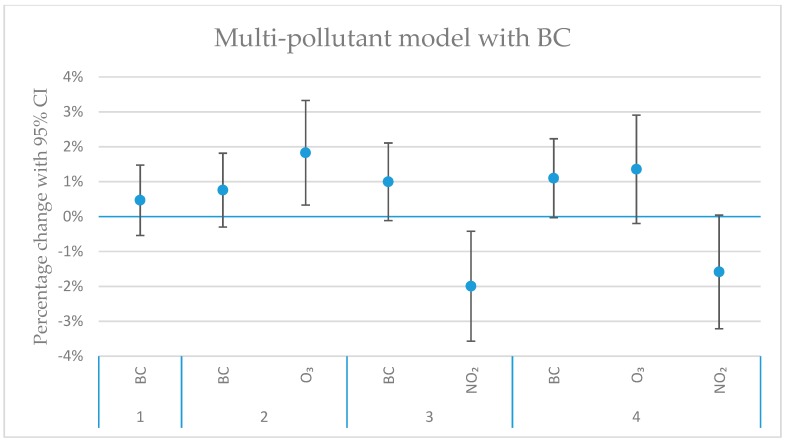
The estimated change in daily mortality (with 95% CI) for an IQR increase in concentration (lag02) of BC, O_3_ and NO_2_ in a multi-pollutant model. Model 1 (furthest to the left) represents the single-pollutant estimate of BC. Model 2 and 3 represent two-pollutant models, with O_3_ (Model 2), and with NO_2_ (Model 3). In Model 4, both BC, O_3_ and NO_2_ are included.

**Figure 7 ijerph-16-01028-f007:**
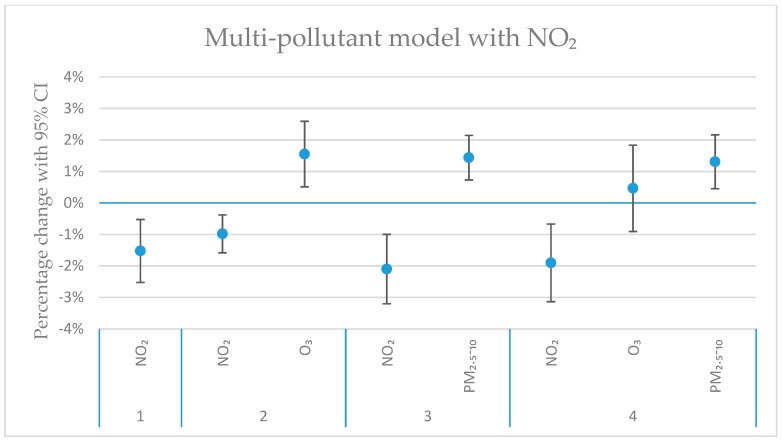
The estimated change in daily mortality (with 95% CI) for an IQR increase in concentration (lag02) of NO_2_, O_3_ and PM_2.5–10_ in a multi-pollutant model. Model 1 (furthest to the left) represents the single-pollutant estimate of NO_2_. Model 2 and 3 represent two-pollutant models, with O_3_ (Model 2), and with PM_2.5–10_ (Model 3). In Model 4, both NO_2_, O_3_ and PM_2.5–10_ are included.

**Table 1 ijerph-16-01028-t001:** Summary statistics of the daily data from 2000–2016 that were used for the study.

Variable	Mean (IQR)	#Days (% Valid Data)
Mortality (N per day)	18.5 (15.2)	6210 (100%)
Maximum temperature (°C)	11.4 (15.0)	6210 (100%)
O_3_ (daily maximum 8-h mean) (µg m^−3^)	51.2 (25.2)	6133 (99%)
PM_2.5_ (µg m^−3^)	6.5 (4.8)	5358 (86%)
PM_10_ (µg m^−3^)	14.5 (8.7)	5999 (97%)
PM_2.5–10_ (µg m^−3^)	8.0 (5.5)	5352 (86%)
NO_2_ (µg m^−3^)	14.4 (9.9)	6101 (98%)
BC (µg m^−3^)	0.6 (0.5)	3316 (53%)
PNC_4_ nm (cm^−3^)	6793 (3484)	2727 (44%)
PNC_7_ nm (cm^−3^)	8701 (4997)	1860 (30%)
PNC^#^ nm (cm^−3^)	9177 (5354)	4328 (70%)

^#^ Calculated PNC_4_ based on regression of PNC_4_ against PNC_7_ (Figure A1).

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
