# Peer review of "Association between Mortality and Short-Term Exposure to Particles, Ozone and Nitrogen Dioxide in Stockholm, Sweden"

_ijerph, 2019, doi:10.3390/ijerph16061028_

Round 1

Reviewer 1 Report

The authors have improved on their original submission. I have the following comments:

1. I would suggest changing the phrasing of lag 01 lag 02 to to 1-day and 2-day lag or include day after lag 01 day to the time is evident in each mention. 

2. This sentence is too vague "The positive, but in most cases, non-significant results associated with exposure to UFP in the above mentioned studies makes it uncertain when it comes to establishing certain conclusions." Need to be specific, for example previous research has not conclusively established a link between short term UFP exposure and mortality.

3. Watch the verb tense in pag3 line 95-96, current phrasing makes it seem like you're referring to a previous study. Say instead We examined the associations between daily mortality and pollutant data obtained from an urban background monitoring station....

4. Figures 5-8 please label your y-axes, include the IQR values for each pollutant in the figure itself or in the legend. No need to say you "separated the effects", it should be clear to readers what a multi pollutant model is. Please make sure your figure fonts match the manuscript font and that make sure the labels have the correct subscripts (PM2.5 vs PM2.5).

5. Page 9 line 247 I would not use more or less use relatively good

6. Page 10 line 318, what do you mean by strange? Do you mean the results are no in line with most published data?

7. Conclusions: I would avoid using excessively certain language like the main reason. You can say the reason we see these associations between some and not other pollutants is potentially due to behavioral patterns. 

Reviewer 2 Report

This resubmission has addressed most of my previous comments. However, there are several minor issues that should be addressed before getting published.

1. As the previous Figure 1 has been moved to supplementary material, please re-number all the remaining figures. 

2. In the sensitivity analysis, was 8 df used in the smooth function for time trend or temperature? Please show the results of the sensitivity analysis using 8df/year for time trend instead of 5 df/year in the supplementary material. 

3. In multi-pollutant models, why not showing the results with PM2.5 and PN4?

4. In abstract, what is the main conclusion of this study?

5. The consistent significant negative association between NO2 and daily mortality should be discussed in more detail. As a lot of previous studies found a significant positive association between NO2 and daily mortality, this contrasting evidence should be interpreted carefully, especially when including NO2 in the title of this paper. 

Round 2

Reviewer 1 Report

Minor comment: 

Figure A1, there seem to be some formatting errors

Author Response

We have changed Figure A1. PMcoarse is changed to PM2.5-10, and PN is changed to PNC. The figure text is also changed, and highlighted in yellow, according to the following: Figure A1. “Daily mean concentrations of PM10, PM2.5, PM2.5-10, BC, O3, NOx and NO2 (µg m-3), and total particle number count (PNC) (>4 nm, cm-3), and the daily number of deaths in Stockholm during the period 2000–2016”.

This manuscript is a resubmission of an earlier submission. The following is a list of the peer review reports and author responses from that submission.

Round 1

Reviewer 1 Report

The authors present an investigation about the association between daily mortality and several pollutants in Stockholm. I have the following comments

Abstract

1.       I don’t think it’s necessary to include the correlation coefficients

Introduction

1.       The authors cite very dated articles in their introduction (over 10 years old). I think their references should be updated to reflect the more recent literature.

2.       The authors’ title focuses on particulates, but the first paragraph of the introduction focuses on NO2 and how robust the associations to health outcomes are after control for other pollutants. It seems to me a framing that should be dropped since it’s not really how the discussion is framed.

3.       In page 2 line 46 I would not use the word cause.

4.       Page 2 line 62, within a 95 CI is weird phrasing. Can just say no significant associations.

5.       Page 2 line 79, I would use understudied rather than underexploited.

Materials and Methods

1.       The authors need to better describe their outcome data. Do deaths from natural causes mean non-accidental deaths? What ICD codes are used?

2.       I don’t think it’s necessary to have a separate table for the pollutant measurement methods

3.       Do the authors have any data on age and/or sex-specific mortality? Would be important to investigate these too.

Results

1.       I don’t think figure 1 is very informative.

2.       Is there a particular reason why 10 ug/m-3 was used for all pollutants. It seems higher than most IQRs and way outside the range of what you would see for BC.

3.       Figure 5 needs to be fixed so the BC estimate is not cut off and you can see the 95% CI bars

Discussion

1.       Authors need to at least briefly discuss what is the mechanism through which air pollution and mortality would be associated.

2.       Page 8 lines 227-231. I’m not sure how this paragraph tracks. The authors start by talking about the negative NO2 association but then move on to PM2.5 and coarse PM.

3.       The authors should discuss how the use of one central measurement at 20 m may lead to exposure misclassification.  

Reviewer 2 Report

This paper addressed an important topic of the ultrafine particle effects on daily mortality in Stockholm using data from 2000 to 2016. In general, this paper is not well written; the main aim is not very clear, the text and methodology need to be further improved.

Major comments

1. The main objective of this study is not very clear, at least reading from the introduction and results. The authors mentioned that they want to compare the PN and BC results with other pollutants, but this was barely done in the manuscript.

2. One of the main concern is using PN4 and PN7, as PN under 10nm generally has a lot of measurement error. The authors should consider using PN above 10nm.

3. Another concern is the simple linear regression approach to get the complete time-series of PN4 based on the PN4-PN7 relationship. Although the model fit may be sufficient, what about the model accuracy (e.g., RMSE)? As the authors are using a time-series design, the daily variation of PN is the key to estimate its effect on mortality. However, daily variation of PN4 may be greatly influenced by using this linear regression. A sensitivity analysis using the real measured PN4 is thus needed.

4. The introduction section needs to be re-written. If the main focus is on PN and BC, why put so much effort describing other well-established air pollutants like NO2 and O3 in the very long first paragraph?

5. In methods, at what point AIC was used? The confounder model seems to be chosen a priori. Also, there are also corresponding model fit criteria for over-dispersed Poisson regression (e.g., QAIC and GCV), so there is no need to change the model. The quasi-Poisson approach is almost always preferable for time-series analysis.

6. For the main model, how many dfs were used for maximum temperature and snowfall? Why using snowfall as confounder? How many lag days are used for the maximum temperature and snowfall?

7. The two-pollutant models are no done based on the correlation? Why only with ozone?

8. There are no sensitivity analyses done/presented. E.g. the authors should check the linearity of the exposure-response functions for all pollutants, including log(population) as offset (as they look at a 17-year period), and changing modeling parameters like different dfs for time trend, temperature, and snowfall and different time lags for temperature (e.g., 2-3 weeks lag for cold effect).

9. Figure 5. Why is the confidence interval for BC that large? Why do the authors only show selected lags, differing for the pollutants? The authors need to show other lags as well.

10. The discussion should also be re-structured, starting with a short summary, and include a paragraph on strengths and limitations.

11. The explanation of insignificant PN effects due to using a centrally located measuring station may be not valid/enough. Measurement errors of using linearly interpolated PN4 and insufficient model adjustments (e.g., the significant negative NO2 effects may be due to inaccurate modeling parameters) could also lead to the nonsignificant finding. Also, the daily variations between different measuring stations across a city may be highly correlated (Cyrys et al. 2008).

12. Conclusion: PM10 was also borderline and could be mentioned.

Reference

Cyrys J, Pitz M, Heinrich J, Wichmann H-E, Peters A. 2008. Spatial and temporal variation of particle number concentration in Augsburg, Germany. Sci Total Environ 401:168-175.

Minor comments

1. The text should be written consistently on past tense.

2. The abbreviations need to be explained at first use (e.g., PN in the abstract is not explained at all in the abstract, it is explained on page 3).

3. The Lu et al. citation in the introduction summarizes only across Chinese populations.

4. Second page: line 56/57: One cannot say that there are only a few studies on UFP and health outcomes. Maybe for specific outcomes.

5. How is PNC different from PN?

6. Second page, line 94: the different BC instruments have been compared, but what was the result of that comparison?

7. Table 1 could include the time-periods covered and could go into the Appendix. Also, which metric is used for ozone? Daily 24-h average or maximum 8-h average?

8. Third page, line 119-11. When pollutant data is missing, this day does not go into the analysis usually.

9. Figure 1 could go into the Appendix.

10. Tables: one decimal is enough for the first column.

11. Page 6, line 167: Is PN meant or PN4?

12. Typo on page 6, line 168: single

13. Figure 4: the correlation coefficients are 0.81 etc. So somewhere needs to be mentioned that the ‘0.__’ is dropped.

14. Page 3, line 174: RR is 1.008 and not 0.8%, the 0.8% is the excess risk, right?

15. Figure 5: why is the CI of BC so large compared to the other pollutants? If the authors want to compare the effects among different pollutants, they should use the IQR instead of 10 µg/m3.

16. Page 8, line 30: There needs to be more discussion on the coarse PM effects. The title of the paper emphasizes coarse particles!

17. Page 8, line 242: I am not sure if ozone largely originates from long-distance transports, especially in summer months.